# The nitrogen removal characterization and ecological risk assessment of *Bacillus* sp. isolated from mariculture systems in China with spatiotemporal difference

**Qian Zhang**[1,2], **Yingeng Wang**[2,3]*, **Zheng Zhang**[2,3], **Yongxiang Yu**[2,3]*, **Chunyuan Wang**[2], **Meijie Liao**[2,3], **Xiaojun Rong**[2,3], **Zhiqi Zhang**[2], **Bin Li**[2,3], **Jianlong Ge**[2,3], **Jinjin Wang**[2]

**1** School of Marine Science and Engineering, Qingdao Agricultural University, Qingdao, Shandong, China, **2** State Key Laboratory of Mariculture Biobreeding and Sustainable Goods, Yellow Sea Fisheries Research Institute, Chinese Academic of Fishery Sciences, Qingdao, Shandong, China, **3** Laboratory for Marine Fisheries Science and Food Production Processes, Lao Shan Laboratory Qingdao, Shandong, China

* wangyg@ysfri.ac.cn (YW); yuyx@ysfri.ac.cn (YY).

## Abstract

The accumulation of nitrogen compounds may worsen the aquatic environment and cause serious economic losses in the aquaculture industry. In this study, the denitrification performance and ecological safety of 120 *Bacillus* sp. isolates with spatial and temporal differences were evaluated based on the aspects of hemolysis, drug resistance, denitrification performance, and purification effect for mariculture wastewater. Firstly, 55/120 safe strains with no hemolytic activity were detected through hemolysis testing. Then, based on selective denitrification medium and colorimetric reagent method, 34/55 *Bacillus* sp. with denitrification effect were screened. For these 34 *Bacillus* sp. isolates, the drug resistance phenotype and genotype, denitrification genes, and enzyme activities related to the nitrogen metabolizing (AMO, HAO, NAR, NIR) were examined. And the MARI was 0.00-0.25, with a multi-drug resistance rate of 17.6%. The drug resistance genes *tetB*, *blaTEM*, and *cfr* and the denitrification genes *nap*, *nor*, and *narG* were detected. Ultimately, 27/34 strains with denitrification function and ecological safety were obtained. In addition, eight *Bacillus* sp. showed certain denitrification effects on nitrogen-containing wastewater treatment. Among them, *B. subtilis* B24 has outstanding denitrification ability, with removal rates of 92%, 62%, 68%, and 30% for $NH_4^+$-N, $NO_2^-$-N, $NO_3^-$-N, and TN in simulated wastewater, respectively. It also has a good denitrification effect in practical applications. This study provides candidate bacterial strains for the treatment of mariculture wastewater.

## Introduction

The aquaculture industry has experienced significant development and changes in the world. Aquaculture production reached a record 130.9 million tons, and Asia contributed 91.4% of the total output [1]. China is the largest aquaculture producer, consumer, and exporter globally; the aquaculture area and output represent over 60% of the total global data and

**Data availability statement:** All relevant data are within the paper and its Supporting Information files.

**Funding:** This study was supported by the National Key Research and Development Program of China in the form of a grant awarded to YXY (2023YFD2402200) and the National Key Research and Development Program of China in the form of a salary for YXY. This study was also supported by the Key Technology Research and Industrialization Demonstration Project of Qingdao City in the form of a grant awarded to ZZ (23-1-3-hygg-22-hy) and the Key Technology Research and Industrialization Demonstration Project of Qingdao City in the form of a salary for ZZ. This study was also supported by the Central Public-interest Scientific Institution Basal Research Fund in the form of a grant awarded to XJR (2023TD29) and the Central Public-interest Scientific Institution Basal Research in the form of a salary for XJR. The specific roles of these authors are articulated in the 'author contributions' section. The funders had no role in study design, data collection and analysis, decision to publish, or preparation of the manuscript.

**Competing interests:** The authors have declared that no competing interests exist.

contributed 1/3 of aquatic products in the world [2]. The rapid development of China's aquaculture industry not only promotes the growing demand for aquatic products but also meets the ecological environment deterioration crisis [3].

The aquaculture industry is facing various environmental challenges, particularly water eutrophication and pollution. These issues impede the sustainable growth of aquaculture and pose heightened risks to human health and environmental safety [4]. In the process of aquaculture, the excessive feeding of bait and the decomposition of animal residues, which results in the production of nitrogen, phosphorus salts, and organic matter, then induces the ecological environment to deteriorate [5]. In particular, toxic metabolites such as $NH_4^+$-N and $NO_2^-$-N, produced through underutilized high-protein feeds, waste, and manure, can seriously affect the quality of aquatic products, posing risks to human health and leading to significant economic losses [6]. The main sources of nitrogen pollution are $NH_4^+$-N, $NO_2^-$-N, and $NO_3^-$-N [7]. Nitrogen content is a critical indicator for assessing water quality. Furthermore, the continuous increase of nitrogen in water may lead to eutrophication or degradation of water quality and sediment, and spreading the disease pathogens.

The traditional method for improve water quality such as water exchange and chemical antiseptics sprinkle can cause ecological and microbial community disturbances [8]. In addition, the abuse of antibiotics and chemical drugs increases the risk of bacterial resistance and even secondary pollution, posing a great threat to aquaculture environmental safety [9,10]. As an environmental friendly and sustainable method, biodegradation is increasingly used for nitrogen removal. Its advantage lies in using microbial metabolic activities to degrade pollutants [11].

*Bacillus* sp. isolates, commonly used as probiotics in aquaculture, promote the growth of aquatic animals, prevent diseases, and improve water quality. Due to their advantages of safety, environmental friendliness, high efficiency, and strong adaptability to the environment, *Bacillus* sp. show great application potential for mariculture wastewater improvement [12,13]. For example, *B. litoralis* strain N31 and *B. subtilis* H1 can effectively remove nitrogen pollutants ($NH_4^+$-N, $NO_2^-$-N, and $NO_3^-$-N) from aquaculture water [13,14]. However, as an ubiquitous biological group in marine environment, the spread of antibiotic resistance and virulence genes has become a consensus ecological risk prevention for microorganism [15,16]. The ecological and biosafety evaluations for *Bacillus* sp. are fundamental requirements for the development of biological products.

However, most current research focuses on the nitrogen removal capabilities of single strains, primarily emphasizing nitrogen removal efficiency under laboratory conditions while lacking systematic assessments of the ecological safety of these strains (such as hemolytic activity, drug resistance, etc.). Additionally, comparative studies on the nitrogen removal performance and ecological safety of strains under different spatial and temporal contexts are relatively scarce, leading to insufficient understanding of the stability and reliability of strains in practical applications. Therefore, this study aims to evaluate and screen the ecological safety and nitrogen removal functions of 120 *Bacillus* sp. isolates with temporal and spatial differences. In terms of ecological safety, the study comprehensively examines the hemolytic activity, drug resistance, and pathogenicity of *Bacillus* strains; regarding nitrogen removal performance, it first uses selective media and colorimetric reagents to preliminarily screen strains with nitrogen removal potential, further analyzing the expression levels of key denitrifying enzymes and the amplification of functional genes. Finally, the nitrogen removal performance of the selected strains is analyzed in simulated wastewater and actual wastewater treatment. This study is expected to provide safe candidate strains for the efficient removal of nitrogen pollutants in aquaculture.

## Materials and methods

### Microorganism and culture medium

The 120 *Bacillus* sp. isolates used in this study were obtained from marine aquaculture systems in various regions of China between 2016 and 2022 and were preserved in the laboratory.

TSB solid medium (Luqiao, Beijing, China) is used for bacterial proliferation culture. Selective media include ammonia nitrogen-degrading bacteria screening medium (NDM) [17], heterotrophic nitrifying medium (HNM), heterotrophic denitrifying medium (HDM), and denitrifying medium (DM-1) [18], and bromothymol blue medium (BTBM) [19], were used for the preliminary screening of denitrifying functional strains. The denitrifying medium (DM-2) is used as the basic culture medium for subsequent reagent colorimetric screening of strains [19]. Inorganic nitrogen degradation test medium (INDTM) [20] is mainly used to detect the denitrification characteristics of the strain in the mixed nitrogen source. The specific components of the culture medium (NDM, HNM, HDM, DM-1, BTBM, DM-2, and INDTM) are shown in S1 Table.

### Hemolytic analysis

5% sterile defibrillated sheep blood was added in TSB agar medium to make blood plate. The 120 *Bacillus* sp. isolates were purified and re-suspended in sterile 1.5% NaCl solution to prepare $2 \times 10^8$ CFU/mL bacterial suspension. Then 100 μL bacterial suspension was inoculated on blood plate and cultured in 28°C for 48 h. Measured the hemolysis circle around the colony, and the strain without hemolysis circle were selected for the next step.

### Preliminary screening of denitrifying *Bacillus* sp.

According to the above hemolytic detection of *Bacillus* sp. isolates, 55 strains of non-hemolytic *Bacillus* sp. isolates were inoculated onto TSB agar medium. After 36 h of purification culture at a constant temperature of 28°C, single colonies were inoculated onto five selective media (NDM, HNM, HDM, DM-1, BTBM) and cultured at a constant temperature of 28°C for 36 h. Strains that could grow on all five media and exhibited a blue halo phenomenon on BTBM were selected for preservation for subsequent experiments. Then, the *Bacillus* sp. isolates initially screened from the above five selective media were streaked for culture and inoculated into 30 mL of liquid DM-2, placed in a shaking incubator at 28°C and 180 r/min for 48 h. A small amount of bacterial liquid was then taken and placed into two 1.5 mL centrifuge tubes, and colorimetric detection was performed using a nitrate reduction reagent kit (Haibo, Qingdao, China) and diphenylamine reagent (Yuanye, Shanghai, China), then observed the color changes. Three parallels were set for each strain, the un-inoculated DM-2 were used as blank control. Detailed information on the isolation of 55 *Bacillus* sp. is shown in S2 Table.

### Drug sensitivity test of the denitrification *Bacillus* sp.

According to the guidelines of the National Committee for Clinical Laboratory Standards, K-B disk diffusion method was used to determine the antibiotic resistance of the above screened denitrification *Bacillus* sp. isolates. A total of 16 antibiotics were selected, including β-lactams (sulbactam, cephalexin, ampicillin, thiamphenicol), quinolones (enrofloxacin, ofloxacin, flumequine), sulfonamides (Sulfamethoxazole, sulfadiazine), macrolides (erythromycin, azithromycin), tetracyclines (tetracycline, doxycycline), aminoglycosides (kanamycin, neomycin), chloramphenicol (florfenicol). *Escherichia coli* ATCC 25922 was utilized as the quality control bacterium. Based on the diameter of the inhibition zone, drug resistance was categorized as sensitive (S), intermediate (I) or resistant (R). Additionally, if the *Bacillus* sp. isolates

exhibit resistance to three or more antibiotics, they are classified as multiple antibiotic resistance (MAR) strains. The multiantibiotic resistance indexes (MARIs) for different strains were calculated according to relevant research methods, which involves determining the ratio of the number of antibiotics resistant to 16 test antibiotics to the total number of antibiotics tested [21].

### Drug resistance genes detected in denitrification *Bacillus* sp.

PCR was utilized to detect the presence of drug resistance genes in the aforementioned *Bacillus* sp. isolates that demonstrate denitrification capabilities. There were 14 resistance genes in 7 categories, including tetracycline resistance genes (*tetA*, *tetB*), extended-spectrum β-lactamases genes (*blaTEM*, *ampC*), aminoglycoside resistance genes [*ant*(3)-*Ia*(*aadA*), *aph*(6')-*Ib*(*strB*)], sulphonamide resistance genes (*sul1*, *sul2*), macrolides genes (*ermA*, *ermX*), chloramphenicol resistance genes (*floR*, *cfr*) and quinolone resistance genes (*qnrA*, *qnrB*). Primers information was shown in S3 Table. The PCR system consisted of 1 μL DNA template, 12.5 μL Taq Master Mix, 1μL upstream primer, 1μL downstream primer, and distilled water with a total volume of 25 μL. The PCR products were analyzed by 1% (w/v) agarose gel electrophoresis and observed under UV light.

### Amplification of denitrification related genes

The functional genes involved in nitrogen metabolism processes, including *nor*, *nap*, and *narG* genes, were amplified from *Bacillus* sp. isolates DNA using conventional PCR. The primer sequences can be found in S3 Table. The PCR system consisted of 1 μL DNA template, 12.5 μL Taq Master Mix, 1 μL upstream primer, 1 μL downstream primer, and double distilled water with a total volume of 25 μL. PCR amplification products were sequenced after 1% agarose gel electrophoresis was used to detect the carrying status of each gene.

### Enzyme activity assay

Inoculate the activated 27 Bacillus sp. into TSB liquid medium, culture for 16 h, and collect the bacterial solution by centrifugation (6000 rpm, 4℃, 10 min). Crush the bacterial cells with ultrasound (300W, 20 min) and collect the supernatant as the crude enzyme solution. The activity of HAO, NAR, NIR, and AMO in the crude enzyme solution was detected by using the hydroxylamine oxidoreductase (HAO) kit, nitrate reductase (NAR) kit, nitrite reductase (NIR) kit, and ammonia monooxygenase (AMO) kit (mlbio, Shanghai, China). The protein content was detected using the BCA method (mlbio, Shanghai, China). Each group was repeated three times, and no enzyme extract was added to the control group. The enzyme activity unit (U) was defined as the amount of enzyme required to catalyze 1 μmol substrate per minute. Enzyme specific activity (U/mg) was defined as enzyme activity divided by protein content in crude enzyme solution.

### Assessment of nitrogen removal capacity

**Denitrification performance of *Bacillus* sp. in simulated wastewater.** A total of 27 *Bacillus* sp. isolates were investigated, the bacterial suspension was inoculated into the inorganic nitrogen degradation test medium after being suspended in normal saline. The concentration of bacteria in the degradation solution was maintained at $(1-2) \times 10^7$ CFU/mL. Basic degradation solution without added bacterial solution and bacterial solutions were used as control groups and experimental groups, respectively. Each group underwent three replicates. The oscillation culture conditions set to maintain 28°C while agitating at 160 rpm. $OD_{600}$ values as well as the concentrations of $NH_4^+$-N, $NO_2^-$-N, $NO_3^-$-N, total nitrogen (TN) in the degradation test solution

were measured every 24 h for 5 d. The inorganic nitrogen remove ability were compared, and strains with higher removal rates were screened for subsequent research.

**Denitrification performance of *Bacillus* sp. in mariculture wastewater.** Eight *Bacillus* sp. isolates with good nitrogen removal capabilities, selected from inorganic nitrogen degradation ability test medium, were applied to mariculture wastewater. The aquaculture water was sourced from laboratory aquarium (*Penaeus vannamei* and *Epinephelus* culture system). Three parallels were set up for both the experimental group and the control group. A total of 540 L of wastewater was divided into 27 aquaculture buckets, each containing 20 L of water. The experimental group was inoculated with bacterial suspension once at a ratio of 1% (v/v) (with a concentration of $1 \times 10^9$ CFU/mL per barrel of bacterial suspension). The control group did not add bacteria. An electric aeration pump was used to aerate the water, maintaining a temperature of 28°C throughout the experiment, which lasted 7 d; other conditions remained consistent, and daily monitoring of $NH_4^+$-N, $NO_2^-$-N, $NO_3^-$-N, temperature, and pH was conducted.

## Determination of the virulence of *B. subtilis* B24

The experiment was conducted using $50 \pm 7.4$ g *Sebastes schlegelii* as the experimental subjects, with 30 fish per bucket. The selected strain B24 was then prepared at concentration gradients of $10^9$ CFU/mL, $10^8$ CFU/mL, $10^7$ CFU/mL for intraperitoneal injection. The control group was injected with an equal amount of sterile liquid TSB. The fish were raised under the same conditions for 7 d, with regular observations of external reactions and mortality rates.

## Analytical methods

$OD_{600}$ and salinity were measured by visible spectrophotometer and salinometer, respectively. $NH_4^+$-N concentration was measured using the indophenol blue spectrophotometric method. The concentration of $NO_2^-$-N was determined using the N-(1-naphthyl)-ethylenediamine dihydro-chloride spectrophotometric method. The concentration of $NO_3^-$-N was determined by the UV spectrophotometric method. The TN concentration was measured using the alkaline potassium persulphate oxidation UV spectrophotometric method [14]. Three replicates were prepared for each group and these statistical results were presented as mean ± standard deviation. Data analysis and graphical representation were completed using Excel, Origin 2021 and GraphPad Prism 9.51 software.

## Results

### Determination of the hemolytic activity of *Bacillus* sp.

The hemolysis test of 120 *Bacillus* sp. isolates with typical spatial and temporal differences was carried out by the blood plate detection method to evaluate its safety. The results indicated that 65 *Bacillus* strains exhibited hemolytic rings in the blood plate; the remaining 55 *Bacillus* sp. isolates did not exhibit hemolytic phenomenon. It was preliminarily inferred that these 55 *Bacillus* sp. isolates posed no pathogenic risk. The types of hemolysis were classified as hemolytic and non- hemolytic based on the size of the hemolytic circle diameter. The diameter of the hemolytic circle is greater than 0 mm for hemolytic and no hemolytic circle for non-hemolytic. Fig 1 shows a schematic diagram of the hemolytic and non- hemolytic of *Bacillus* sp. isolates.

### Preliminary screening of denitrifying *Bacillus* sp.

First, 55 non-hemolytic *Bacillus* sp. isolates were cultured using five selective media (DNM, HNM, HDM, DM-1, BTBM) to observe their proliferation trends and color development

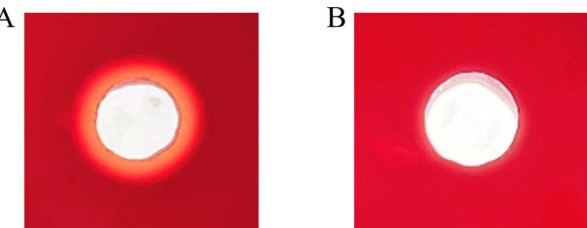

**Fig 1. A diagram indicating the hemolytic and non-hemolytic activities of certain *Bacillus* sp. isolates.** (A) represents the hemolytic diagram of the strain. (B) represents the non-hemolytic diagram of the strain.

for preliminary screening. The results showed that, 47/55 strains grew well on DNM; 38/55 strains could grow on HNM, HDM, and DM-1; 44/55 strains exhibited blue halo phenomena on BTBM. Based on the growth conditions and color development of the above *Bacillus* sp. isolates, 34 *Bacillus* sp. isolates were initially screened for their nitrogen removal function. Then, a nitrate reduction kit and diphenylamine colorimetric method were used for further screening of the 34 *Bacillus* sp. isolates. The colorimetric results indicated that the nitrate reduction kit detected all 34 *Bacillus* sp. isolates as positive, and after adding diphenylamine reagent, none of the 34 *Bacillus* sp. isolates changed color, indicated that all 34 strains had denitrification activity. In summary, a total of 34 *Bacillus* sp. isolates with nitrogen removal function were finally selected.

## Drug sensitivity test

The drug sensitivity test revealed that 34 *Bacillus* sp. isolates exhibited the highest resistance rate to sulfadiazine, with 23 out of 34 showing resistance (Fig 2). But sensitive to florfenicol (34/34), flumequine (34/34), sulbactam (34/34), doxycycline (34/34), cephalexin (33/34),

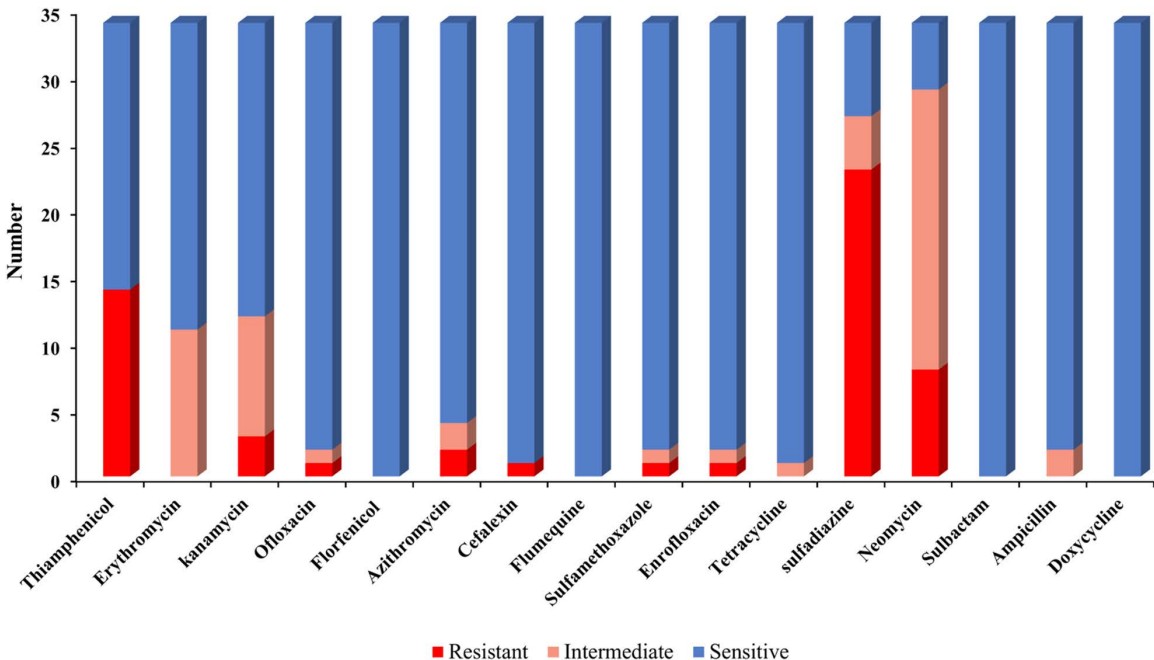

**Fig 2. Drug sensitivity analysis of 34 *Bacillus* sp. isolates to 16 antibiotics.**

tetracycline (33/34), sulfamethoxazole (32/34), enrofloxacin (32/34), ampicillin (32/34) and ofloxacin (32/34). The sensitivity of 34 *Bacillus* sp. isolates to kanamycin (22/34), thiamphenicol (20/34) and neomycin (5/34) was low, MARI were 0.00-0.25, with 6 strains resistant to 3 or more antibiotics, resulting in a multi-drug resistance rate of 17.6%.

## Detection of drug resistance genes

The PCR amplification results showed that, 14 drug resistance genes were used and only 3 genes were detected. Among which, 34 strains (100%) were contain tetracycline gene *tetB*, one strain (2.94%) contain chloramphenicol gene *cfr*, and one strain (2.94%) contain β-lactam gene *blaTEM* (Fig 3). Quinolone resistance genes *qnrA*, *qnrB*, chloramphenicol resistance gene *flor*, tetracycline resistance gene *tetA*, aminoglycoside resistance genes *ant(3')-Ia(aadA)* and *aph(6')-Id(strB)*, sulfonamide resistance genes *sul1*, *sul2*, macrolide resistance genes *ermA* and *ermX* were not detected.

Among the 34 *Bacillus* sp. isolates, no resistance gene was found in the sulfadiazine-resistant strains. Furthermore, certain *Bacillus* sp. isolates exhibited sensitivity to tetracycline, cephalexin, sulbactam, thiamphenicol and florfenicol. but the genes of *tetB*, *blaTEM* and *cfr* were identified.

## Identification of nitrogen metabolism related functional genes

By using PCR methods to amplify the relevant functional genes involved in the denitrification process from 34 *Bacillus* sp. isolates. The results showed that, 27/34 *Bacillus* sp. isolates

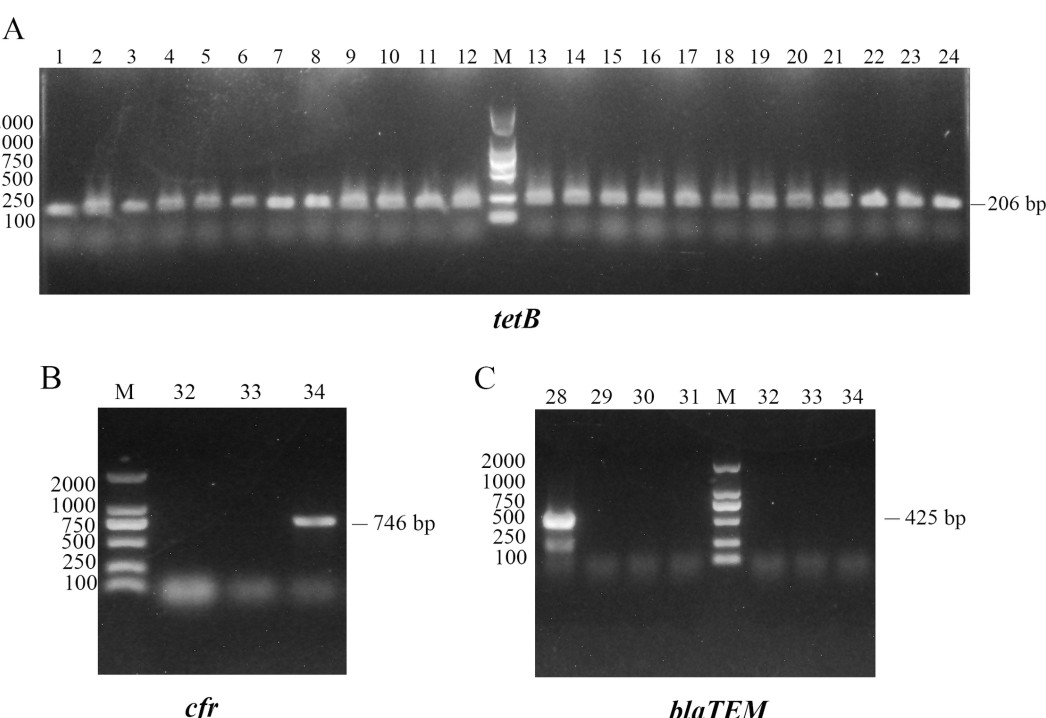

**Fig 3.  PCR amplification of antibiotic resistance genes in denitrifying *Bacillus* sp. isolates.** (A) electrophoresis schematic diagram of *tetB* gene, lane M indicates the molecular marker (2000 bp), lanes 1-24 indicate the *tetB* positive strains (206 bp). (B) electrophoresis schematic diagram of *cfr* gene, lane M indicates the molecular marker (2000 bp), lane 34 indicates the *cfr* positive strain (746 bp). (C) electrophoresis schematic diagram of *blaTEM* gene, lane M indicates the molecular marker (2000 bp), lane 28 indicates the *blaTEM* positive strain (425 bp).

successfully amplified the relevant functional genes *nap*, *nor*, and *narG* involved in the process of denitrogenation and obtained 2040 bp PCR products, 1917 bp and 256 bp, respectively. The PCR amplification results of functional genes involved in the denitrification process are shown in Fig 4.

### Enzyme activity evaluation

The denitrification enzyme activity of 27 *Bacillus* sp. isolates with amplified denitrification functional genes was measured, mainly including HAO, NAR, NIR, and AMO. The measurement results are shown in Table 1. The denitrification process of the 27 *Bacillus* sp. isolates, HAO, NAR, NIR, and AMO all exhibited a certain level of expression, and the HAO specific activity of the 27 strains being the highest ($0.0321 \pm 0.0007$U/mg-$0.0666 \pm 0.0007$U/mg protein).

### Assessment of nitrogen removal capacity

**The nitrogen removal capacity of *Bacillus* sp. in simulated wastewater.** Through the INDTM to simulate mariculture wastewater, the nitrogen removal effects of 27 *Bacillus* sp.,

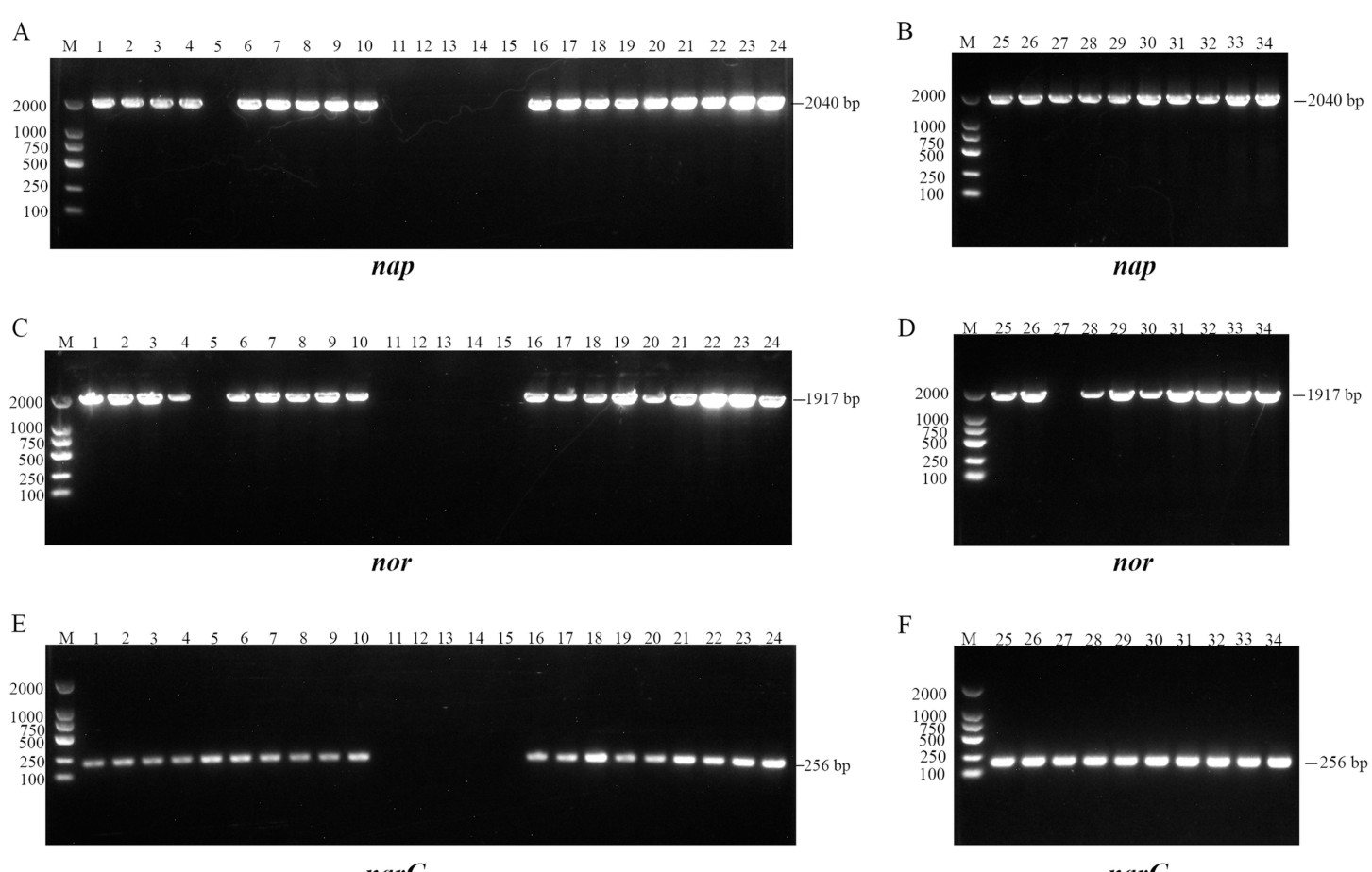

**Fig 4. PCR amplification results of three functional genes from 34 *Bacillus* sp. isolates.** (A and B) Gel electrophoresis of *naP* in 34 *Bacillus* sp. isolates (2040 bp). Lane M: molecular marker (2000 bp). (C and D) Gel electrophoresis of *nor* in 34 *Bacillus* sp. isolates (1917 bp). Lane M: molecular marker (2000 bp). (E and F) Gel electrophoresis of *narG* from 34 *Bacillus* sp. isolates (256 bp). Lane M: molecular marker (2000 bp).

**Table 1. The specific activity of nitrogen removal key enzymes of 27 *Bacillus* sp. isolates.**

| Strain number | Specific activity (U/mg protein) | | | |
| --- | --- | --- | --- | --- |
| | Hydroxylamine oxidoreductase (HAO) | Nitrate reductase (NAR) | Nitrite reductase (NIR) | Ammonia monooxygenase (AMO) |
| B1 | 0.0453 ± 0.0007 | 0.0196 ± 0.0002 | 0.0144 ± 0.0000 | 0.0089 ± 0.0002 |
| B2 | 0.0447 ± 0.0011 | 0.0128 ± 0.0001 | 0.0126 ± 0.0003 | 0.0094 ± 0.0001 |
| B3 | 0.0414 ± 0.0009 | 0.0115 ± 0.0001 | 0.0101 ± 0.0001 | 0.0068 ± 0.0001 |
| B4 | 0.0556 ± 0.0004 | 0.0132 ± 0.0002 | 0.0130 ± 0.0001 | 0.0107 ± 0.0002 |
| B5 | 0.0486 ± 0.0004 | 0.0103 ± 0.0001 | 0.0130 ± 0.0001 | 0.0093 ± 0.0001 |
| B6 | 0.0394 ± 0.0009 | 0.0146 ± 0.0001 | 0.0154 ± 0.0003 | 0.0063 ± 0.0001 |
| B7 | 0.0385 ± 0.0005 | 0.0103 ± 0.0002 | 0.0138 ± 0.0002 | 0.0108 ± 0.0003 |
| B8 | 0.0397 ± 0.0008 | 0.0072 ± 0.0001 | 0.0093 ± 0.0000 | 0.0064 ± 0.0001 |
| B9 | 0.0403 ± 0.0005 | 0.0064 ± 0.0000 | 0.0105 ± 0.0002 | 0.0086 ± 0.0001 |
| B10 | 0.0666 ± 0.0007 | 0.0104 ± 0.0001 | 0.0157 ± 0.0002 | 0.0107 ± 0.0001 |
| B11 | 0.0562 ± 0.0005 | 0.0163 ± 0.0003 | 0.0112 ± 0.0002 | 0.0102 ± 0.0001 |
| B12 | 0.0433 ± 0.0010 | 0.0152 ± 0.0002 | 0.0162 ± 0.0000 | 0.0094 ± 0.0001 |
| B13 | 0.0456 ± 0.0000 | 0.0172 ± 0.0003 | 0.0111 ± 0.0002 | 0.0079 ± 0.0001 |
| B14 | 0.0622 ± 0.0005 | 0.0079 ± 0.0001 | 0.0123 ± 0.0001 | 0.0105 ± 0.0000 |
| B15 | 0.0569 ± 0.0008 | 0.0162 ± 0.0003 | 0.0146 ± 0.0002 | 0.0094 ± 0.0001 |
| B16 | 0.0361 ± 0.0004 | 0.0120 ± 0.0001 | 0.0101 ± 0.0002 | 0.0079 ± 0.0000 |
| B17 | 0.0382 ± 0.0004 | 0.0165 ± 0.0003 | 0.0099 ± 0.0001 | 0.0062 ± 0.0001 |
| B18 | 0.0414 ± 0.0003 | 0.0130 ± 0.0002 | 0.0148 ± 0.0002 | 0.0102 ± 0.0002 |
| B19 | 0.0357 ± 0.0008 | 0.0099 ± 0.0001 | 0.0106 ± 0.0002 | 0.0079 ± 0.0001 |
| B20 | 0.0321 ± 0.0007 | 0.0010 ± 0.0002 | 0.0099 ± 0.0002 | 0.0090 ± 0.0002 |
| B21 | 0.0322 ± 0.0008 | 0.0147 ± 0.0002 | 0.0084 ± 0.0000 | 0.0071 ± 0.0002 |
| B22 | 0.0429 ± 0.0003 | 0.0141 ± 0.0002 | 0.0120 ± 0.0003 | 0.0096 ± 0.0001 |
| B23 | 0.0517 ± 0.0010 | 0.0174 ± 0.0001 | 0.0141 ± 0.0002 | 0.0101 ± 0.0001 |
| B24 | 0.0466 ± 0.0007 | 0.0195 ± 0.0002 | 0.0102 ± 0.0001 | 0.0075 ± 0.0001 |
| B25 | 0.0517 ± 0.0006 | 0.0073 ± 0.0001 | 0.0131 ± 0.0002 | 0.0096 ± 0.0001 |
| B26 | 0.0515 ± 0.0005 | 0.0171 ± 0.0001 | 0.0149 ± 0.0001 | 0.0067 ± 0.0001 |
| B27 | 0.0441 ± 0.0006 | 0.0150 ± 0.0003 | 0.0113 ± 0.0001 | 0.0064 ± 0.0001 |

which amplified the nitrogen removal functional genes, were screened and evaluated. The results showed that, 8/27 *Bacillus* sp. isolates in the experimental group had a certain removal effect on $NH_4^+$-N, $NO_2^-$-N, $NO_3^-$-N, and TN (Fig 5), while the nitrogen degradation capability of the remaining 19 strains was poor (S4 Fig). During the 5d cultivation process, these 8 *Bacillus* sp. isolates maintained a good growth trend, with $NH_4^+$-N removal rates of 80%-92%, $NO_2^-$-N removal rates of 58%-72%, $NO_3^-$-N removal rates of 27%-68%, and TN removal rates of 20%-37%. Among them, strain B24 (*B. subtilis*) showed the best nitrogen removal effect compared to the other seven strains, the removal rate of $NH_4^+$-N were 92%, the removal rate of $NO_2^-$-N were 62%, the removal rate of $NO_3^-$-N were 68%, and the removal rate of TN were 30%. The nitrogen content of the blank control group remained relatively stable.

**Nitrogen removal effect in mariculture wastewater.** Eight *Bacillus* sp. isolates with good nitrogen removal effects were screened from the inorganic nitrogen degradation ability test medium and further applied in mariculture wastewater. As shown in (Fig 6), the initial pH of the aquaculture water was 7.89, and the pH values of the experimental group and the control group gradually increased within 7 d, with the final pH of the experimental group

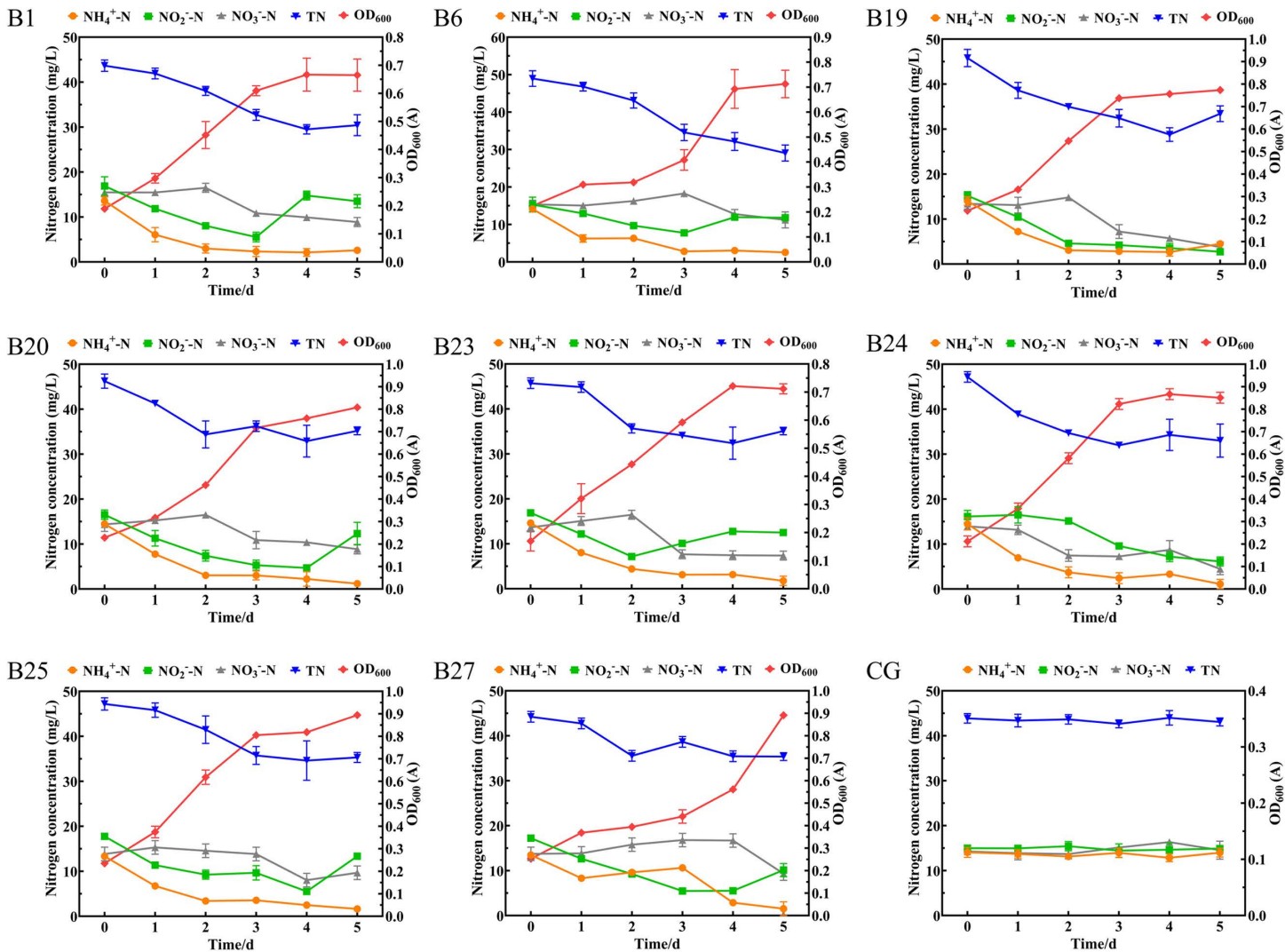

**Fig 5. Removal rates of NH$_4$$^+$-N, NO$_2$$^-$-N, NO$_3$$^-$-N, and TN by eight *Bacillus* sp. isolates in inorganic nitrogen degradation ability test medium.** (B1, B6, B19, B20, B23, B24, B25 and B27) represent the denitrification performance of experimental strains B1, B6, B19, B20, B23, B24, B25 and B27, respectively. (CG) Changes in nitrogen content in the control group.

being 8.03-8.19 and the final pH of the control group being 8.15. The eight *Bacillus* sp. isolates in the experimental group showed certain nitrogen removal effects for NO$_2$$^-$-N and NH$_4$$^+$-N in mariculture wastewater, with 44%-63% and 11%-32% removal rates, respectively. Additionally, four *Bacillus* sp. isolate B1, B20, B24 and B27 exhibited certain removal effects on NO$_3$$^-$-N, with removal rates of 14%-42%. In the blank control group, the concentrations of NO$_2$$^-$-N and NH$_4$$^+$-N in the water showed a slight decrease, with final concentrations of 7.04 mg/L and 1.6 mg/L, respectively.

Among them, strain B24 performed better than the other seven *Bacillus* sp. isolates in the application in seawater aquaculture wastewater. The initial concentration of NH$_4$$^+$-N in the water with added strain B24 was 8.13 mg/L, and during the first 3 d, the NH$_4$$^+$-N concentration maintained a slight fluctuation trend, decreasing to 5.50 mg/L by the 7th day, with a final removal efficiency of 32%. The NO$_2$$^-$-N concentration decreased from the initial concentration of 2.03 mg/L at 0 days to 0.6 mg/L at 0.5 d, reaching the lowest concentration, with a removal

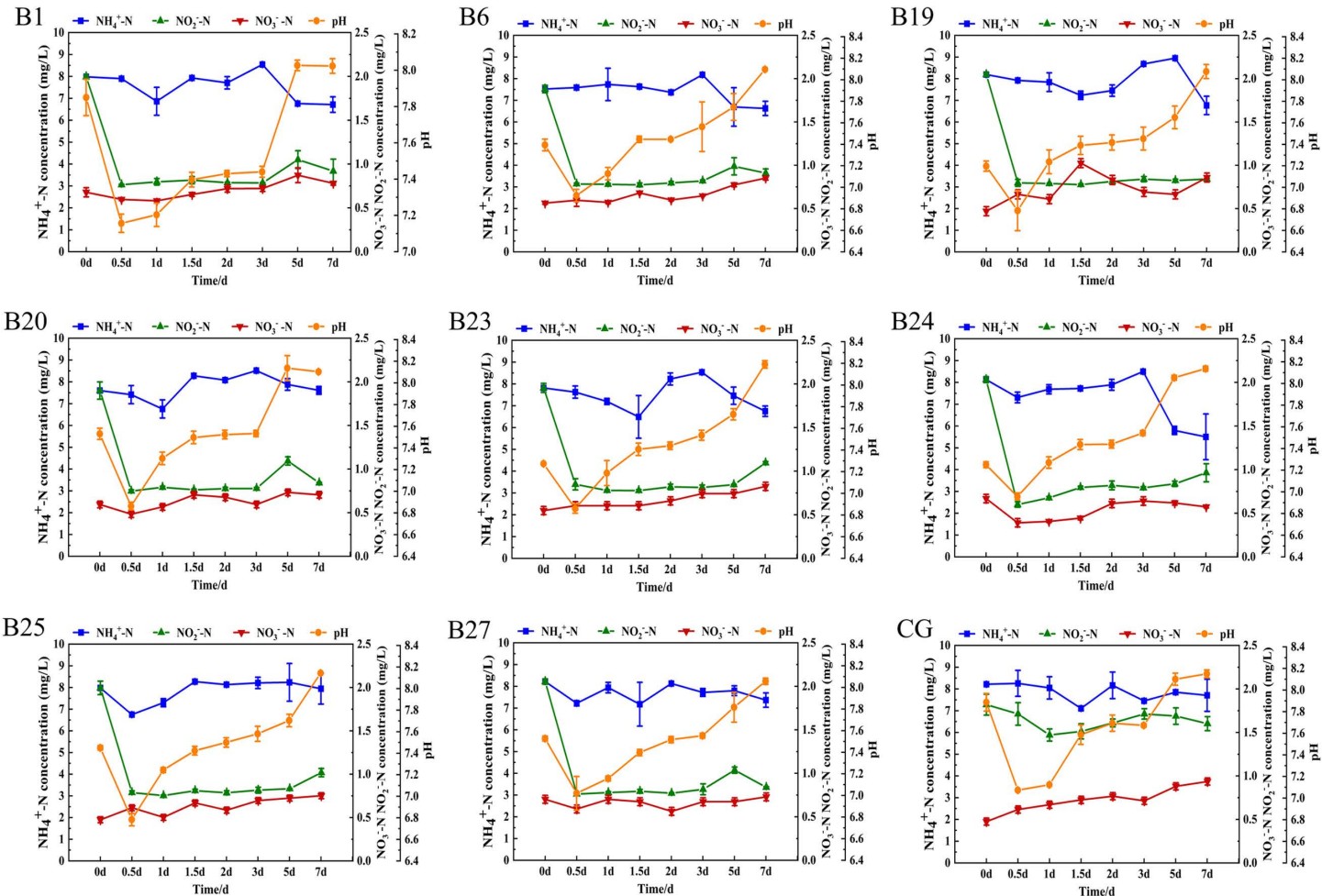

**Fig 6. Application effect of eight *Bacillus* sp. isolates on nitrogen removal in mariculture wastewater.** (B1, B6, B19, B20, B23, B24, B25 and B27) represent the nitrogen removal performance of experimental strains B1, B6, B19, B20, B23, B24, B25 and B27, respectively. (CG) variation of nitrogen content in the control group.

efficiency of 70%. The $NO_3^-$-N concentration decreased from the initial concentration of 0.67 mg/L at 0 days to 0.39 mg/L at 0.5 days, reaching the lowest value, with a removal efficiency of 42%.

### Determination of *B. subtilis* B24 virulence

Based on the regular observations over 7 d, *B. subtilis* B24 at injection concentration gradients of $10^9$ CFU/mL, $10^8$ CFU/mL, and $10^7$ CFU/mL, as well as the control group, showed no mortality in *Sebastes schlegelii*. Additionally, the fish did not exhibit any symptoms of disease such as ulcers, redness, or enteritis, and their vitality was good. This indicates that the strain B24 is non-pathogenic to *Sebastes schlegelii* at concentrations below $10^9$ CFU/mL, demonstrating a certain level of biosafety.

## Discussion

The accumulation of nitrogen pollutants in the water environment will not only disrupt the balance of aquatic ecosystems but also induce the eutrophication of aquaculture water and

impact the health of aquatic animals. In recent years, the biological nitrogen removal process has gradually become the focus due to its efficient, environmentally friendly, and economical application for nitrogen removal. Therefore, this study aims to select the safe and effective *Bacillus* sp. isolates for nitrogen pollution improvement in the mariculture system. And the hemolytic, drug resistance, and denitrification performance of 120 *Bacillus* sp. isolates with typical spatial and temporal differences were screened and evaluated, and then the strains with ecological safety were obtain; the results provided alternative strains for microbial regulation of water quality in mariculture.

Host safety is the basic requirement for aquatic probiotics [22,23]. Hemolytic activity and drug resistance are important indicators of ecological risk assessment [24]. Since hemolysin, as a virulence factor, can cause anemia and edema in the host, non-hemolytic probiotic strains have greater potential application value [25]. Therefore, in this study, 120 *Bacillus* sp. isolates were screened for hemolysis by blood plate to ensure no toxic side effects on cultured organisms. Among them, 65 strains with hemolytic activity were removed for screening, and 55 safe strains without hemolytic activity could be used for subsequent screening studies. Subsequently, combined with the denitrification function screening medium and reagents, 34 *Bacillus* sp. isolates with denitrification function were finally screened.

Antibiotic sensitivity is also one of the most important characteristics of probiotics [24]. Some *Bacillus* sp. may produce drug-resistant strains, increasing the risk of antibiotic resistance gene transmission [26]. Related research has found that *Bacillus* sp. are sensitive to ampicillin, cephalexin, and tetracycline [26,27], which is similar to the results of this study. In this study, over 92% of the *Bacillus* sp. isolates were sensitive to ampicillin, cephalexin, tetracycline, florfenicol, and flumequine, sulbactam, doxycycline, sulfamethoxazole, enrofloxacin, ofloxacin. Nearly 82% of the strains had a MARI value less than 0.13. MARI reflects the degree of environmental pollution that antibiotics may cause to human health. A value higher than 0.2 indicates a high risk of antibiotic exposure, and a value lower than 0.2 indicates a low risk of antibiotic exposure [28]. This study showed that the risk of antibiotic exposure of *Bacillus* sp. isolates were low.

In addition, the transfer of antibiotic resistance genes between bacteria is the key factor leading to the rapid spread of antibiotic resistance, which increases the risk of antibiotic treatment failure and poses a serious threat to public health [29]. In this study, only three of the seven classes of antibiotic resistance genes were detected, with only three drug resistance genes, among which *cfr* (2.94%) and *blaTEM* (2.94%) all had low detection rates except *tetB* (100%), indicating that the risk of resistance gene transmission of resistance genes from these 34 *Bacillus* sp. is low. However, some *Bacillus* sp. showed sensitive phenotypes to tetracycline, cephalexin, ampicillin, florfenicol, and other antibiotics, but the resistance genes *tetB*, *cfr*, and *blaTEM* were still detected. Researchers have observed a difference between the antibiotic resistance phenotype and the prevalence of antibiotic resistance genes, which may be due to the expression of antimicrobial phenotypes under the stimulation of many different genetic determinants [30]. Another possible reason is that the antimicrobial phenotype is not only mediated by resistance genes but also by membrane structure and physiological metabolism [16].

To identify strains that effectively remove nitrogen and to understand their nitrogen metabolism mechanisms, we tested denitrification genes and nitrogen cycle-related enzyme activities in 34 strains of denitrifying *Bacillus*. The results show that 27 out of 34 strains of *Bacillus* successfully amplified the denitrification functional genes *nor*, *nap*, and *narG*, indicating that these strains have denitrification capability. Moreover, the process of microbial removal of nitrogen-containing compounds is closely related to the catalytic action of enzymes [31]. AMO and HAO are key enzymes in the nitrification process, while NAR and

NIR are key enzymes in the denitrification process [32]. Therefore, the activities of four key enzymes (AMO, HAO, NAR, NIR) were detected and indicated that the 27 *Bacillus* sp. isolates had the function of promoting the nitrogen cycle. The results show that the four enzymes AMO, HAO, NAR, and NIR are active in 27 *Bacillus* sp., further confirming the presence of nitrogen conversion pathways in these strains. Among them, AMO oxidizes $NH_4^+$-N to $NH_2OH$, and HAO can oxidize $NH_2OH$ to various final products of denitrification [33]. Although the *amoA* and *hao* were not amplified, the AMO and HAO enzyme activities were expressed to a certain extent, and the expression of AMO enzyme activity was the highest, which indicated that the oxidation process occurred in the strain, and the genome should be investigated. NAP (periplasmic nitrate reductase) and NAR are responsible for the conversion of $NO_3^-$-N to $NO_2^-$-N. NAP and NAR were mainly reduced under aerobic and anaerobic conditions, respectively [34,35]. In this study, the *nap* gene and *narG* gene were successfully amplified, and NAR enzyme activity was detected, which indicated that the reductive reaction existed in the strain. NIR is divided into *nirS* (cytochrome cd1 nitrite reductase) and *nirK* (copper nitrite reductase) [36]. In this study, NIR enzyme activity could be detected, but *nirS* and *nirK* could not be successfully amplified from the strain. This perhaps was explained by the reasons that some of the nitrogen removal reaction was conducted by chemical-catalyzed processes rather than enzyme-catalyzed, or the design of the primer sequences is not applicable to these *Bacillus* strains. Therefore, it is necessary to further study the functional gene analysis and denitrification mechanism of the *Bacillus* sp. isolates. In conclusion, the 27 *Bacillus* sp. not only showed denitrification ability at the gene level but also promoted the nitrogen cycle at the enzyme activity level.

*Bacillus* species, as microbial agents, have garnered significant attention due to their substantial potential for treating nitrogenous waste in aquaculture. Therefore, this study further evaluates the nitrogen removal capabilities of *Bacillus* sp. isolates by simulating the application effects of cultured wastewater in the laboratory and actual mariculture wastewater. The selected strain *B. subtilis* B24 exhibited a high nitrogen removal capacity and has potential applications for improving water quality. Under simulated mixed nitrogen sources, B24 achieved maximum removal rates of 92% for $NH_4^+$-N, 62% for $NO_2^-$-N, and 68% for $NO_3^-$-N. Related studies have found that the removal rates of $NH_4^+$-N and $NO_3^-$-N by B24 is higher than those of the *Bacillus* strains K8, N2, and N5 (*B. subtilis*), HYS (*B. albus*), H4 (*B. amyloliquefaciens*), and S1 (*B. velezensis*) [37]. Additionally, strain B24 showed an overall decrease in nitrogen removal efficiency and an increase in water pH in actual aquaculture wastewater, which may be due to the presence of various ions and complex microbial communities in the actual wastewater [38]. Furthermore, the low nitrogen removal efficiency of *Bacillus* sp. may also be influenced by factors such as pH, temperature, salinity, dissolved oxygen, and nutrients [39]. In summary, *B. subtilis* B24 demonstrates good nitrogen removal potential; its efficiency in practical applications is necessitating further optimization.

## Conclusions

In this study, 55/120 non-hemolytic strains of *Bacillus* sp. were detected from previously isolated strains in marine aquaculture systems, and further screening identified 34/55 strains with denitrification capabilities. The risk assessment for antibiotic resistance of these 34/55 strains showed a MARI of 0.00-0.25, with a multidrug resistance rate of 17.6% and the detection of resistance genes *tetB*, *blaTEM*, and *cfr*. In addition, denitrification functional genes *nap*, *nor*, and *narG* were successfully amplified from 27/34 *Bacillus* sp., and the activities of nitrogen metabolism-related enzymes (AMO, HAO, NAR, NIR) were all expressed. Ultimately, 8/27 *Bacillus* sp. demonstrated good denitrification effects in aquaculture wastewater

applications, with *B. subtilis* B24 showing the best performance, particularly in simulated wastewater, achieving removal rates of 92%, 62%, 68%, and 30% for $NH_4^+$-N, $NO_2^-$-N, $NO_3^-$-N, and TN, respectively. Therefore, this study provides sufficient information for the application of *B. subtilis* B24 in improving water quality in aquaculture.

## Supporting information

**S1 Table. The specific components of the culture medium used in this study.**
(DOC)

**S2 Table. Information of 55 non-hemolytic *Bacillus* sp. isolates .**
(DOC)

**S3 Table. The primers used to this study.**
(DOC)

**S4 Fig. Removal rates of $NH_4^+$-N, $NO_2^-$-N, $NO_3^-$-N, and TN by the remaining 19 *Bacillus* sp. isolates in the INDTM .**
(TIF)

**S1 Raw Images. Raw images .**
(PDF)

**S6 File. Date.**
(XLSX)

## Author contributions

**Conceptualization:** Yingeng Wang.

**Data curation:** Qian Zhang, Yongxiang Yu.

**Formal analysis:** Yongxiang Yu.

**Funding acquisition:** Yingeng Wang, Zheng Zhang, Xiaojun Rong.

**Investigation:** Qian Zhang, Chunyuan Wang.

**Methodology:** Zheng Zhang, Bin Li, Jianlong Ge.

**Project administration:** Chunyuan Wang, Jinjin Wang.

**Resources:** Yongxiang Yu, Meijie Liao, Bin Li.

**Software:** Zhiqi Zhang, Jianlong Ge.

**Supervision:** Meijie Liao, Jinjin Wang.

**Validation:** Chunyuan Wang, Meijie Liao.

**Visualization:** Qian Zhang, Zheng Zhang, Zhiqi Zhang.

**Writing – original draft:** Qian Zhang, Yongxiang Yu.

**Writing – review & editing:** Yingeng Wang, Xiaojun Rong.

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
