## [Decision Letter · Decision Letter 0]

26 Dec 2024

PONE-D-24-56326The nitrogen removal characterization and ecological risk assessment of Bacillus sp. isolated from mariculture systems in China with spatiotemporal differencePLOS ONE

Dear Dr. Yu,

Thank you for submitting your manuscript to PLOS ONE. After careful consideration, we feel that it has merit but does not fully meet PLOS ONE’s publication criteria as it currently stands. Therefore, we invite you to submit a revised version of the manuscript that addresses the points raised during the review process.

We look forward to receiving your revised manuscript.

Kind regards,

Xingbin Sun

Academic Editor

PLOS ONE

Journal Requirements:

5. In the online submission form, you indicated that all data were list in the manuscript, and the row data will be made available on request.

Reviewers' comments:

Reviewer's Responses to Questions

**Comments to the Author**

1. Is the manuscript technically sound, and do the data support the conclusions?

Reviewer #1: Yes

Reviewer #2: Yes

Reviewer #3: Yes

2. Has the statistical analysis been performed appropriately and rigorously? 

Reviewer #1: Yes

Reviewer #2: Yes

Reviewer #3: No

3. Have the authors made all data underlying the findings in their manuscript fully available?

Reviewer #1: Yes

Reviewer #2: Yes

Reviewer #3: Yes

4. Is the manuscript presented in an intelligible fashion and written in standard English?

Reviewer #1: Yes

Reviewer #2: Yes

Reviewer #3: No

5. Review Comments to the Author

Reviewer #1: 1. The abstract is too long and should be further summary and simplification.

2. The focus and novelty of this paper are not prominent.

3. The purpose and signification of the paper in the introduction did not clearly describe and in the last paragraph of the introduction should indicate the content that this paper intends to do, rather than the final experimental results of this paper.

4. In page 4 line 83-86, this sentence has a grammatical error.

5. In line 108, line 146-147, line 154, line 162-163, line 185, line 201, line 211, line 263 and line 270-271, there are some issue with those title.

6. The data in the subsequent experimental results of the paper should not appear in the Materials and methods section.

7. In page 9 line 180-181, the meaning conveyed by this sentence in unclear.

8. The description of the experimental results section is too detailed and should be appropriately simplified. For example, the sentence in line 236-239 can be delete.

9. the content of the discussion section is too scattered and does not highlight the novelty of the paper.

10. The conclusion section should summarize the results of the paper, rather than describing what the paper has done.

11. The nitrogen removal rates of B. subtilis B24 was not high, so where is the potential application of this strain? And why not directly screen for efficient degrading Bacillus from natural environments or aquaculture water bodies?

Reviewer #2: 1. Line 241, 242. Certain isolates exhibited sensitivity to tetracycline, cephalexin, sulbactam, thiamphenicol and florfenicol, but the genes of tetB, blaTEM and cfr were identified. Please explain Why?

2. Line 278-280. Eight isolates maintained a good growth trend, with NH4 +-N removal rates of 80%-92%, NO2 --N removal rates of 58%-72%, NO3 --N removal rates of 27%-68%, while the removal rates of TN were only 20%-37%. Please explain “Why”.

3. In figure 5 and figure 6. It is best to label B1, B6, B19, B20, B23, B24, B25, B27, and control in the figure, respectively.

4. Line 378-380. NIR enzyme activity could be detected, but nirS and nirK could not be successfully amplified from the strain. Please explain “Why”.

Reviewer #3: In this manuscript, the authors comprehensively evaluated the nitrogen removal performance and ecological safety of 120 strains of Bacillus with spatial and temporal differences. Overall, the data was presented in a scientific way and the results and conclusions are also reasonable. However, some revisions need to be done to improve the manuscript. The authors should take the following suggestions seriously and the grammar errors in the paper should be checked and corrected carefully.

1.The paper should be read by a native and some terms should be improved. There are many sentences that need to be rewritten, such as lines 41-42, lines 80-82, lines 90-101, lines 224-225，lines 338-339 and so on.

2.Although 120 strains of Bacillus were used in the study, the details on how these strains adequately represent spatial and temporal differences in mariculture systems of China were lacking. For example, the influence of sample collection standards and culture system types in different regions and time points on strain characteristics was not explained in depth, which may affect the extrapolation of research results.

3.The format of the references in lines 48-49 and lines 343-344 is inconsistent with other references.

4.In the process of discussion, some contents were not logical enough. For example, when describing the safety of Bacillus as an aquatic probiotic, there was a direct jump from hemolysis test to antibiotic sensitivity, and there was a lack of transitional analysis in the middle. The logical correlation between different results should be strengthened to make the discussion more coherent and smooth.

5.The marking of some Figures is not perfect enough. The strain number represented by each curve in Figure 5 is not marked directly, so it needs to be found in the text, which affects the independence and readability of the Figure. It should be marked clearly in the Figure, so that readers can understand the meaning of the Figure without referring to the text.

6.In the part of hemolysis analysis, the description of the experimental operation is clear, but the specific criteria for determining the type of hemolysis (strong, weak, and non-hemolysis) are not clearly defined. For example, the corresponding relationship between the size of the hemolysis circle and the type of hemolysis may lead to the difference in judgment between different experimenters, and detailed criteria should be added.

7.It is found that strains are sensitive to some antibiotics but carry corresponding resistance genes. Can we explore the molecular mechanism, such as gene regulation mechanism and detection of drug-resistance gene expression level, to explain this contradictory phenomenon.

8.The introduction of various media components is too lengthy, and it can be focused on the key components and effects directly related to experimental screening or detection, simplifying the description of conventional components (such as glucose, sodium chloride, etc.).

9.When discussing the safety and drug resistance characteristics of Bacillus as a probiotic, literatures such as [12], [32] and [33] were cited, but only the results of previous studies were briefly mentioned, without in-depth analysis of the specific differences and connections between these literatures and the characteristics of strains and experimental results in this study. For example, literature [12] describes the nitrifying and denitrifying characteristics of a new strain of Bacillus. Although the denitrifying function of Bacillus is involved in this study, the characteristics of the strain in this literature were not compared with the strains screened in this study in terms of drug resistance and ecological safety, which makes the reference seem superficial. The supporting role of cited literature for in-depth discussion of research results is not fully utilized.

10.For some key experimental operations and detection methods (such as enzyme activity detection methods), there is no reference to provide method basis or comparison with existing research methods, which makes the innovation and reliability of experimental methods lack sufficient literature support, and it is difficult to judge the status and applicability of these methods in this field of research.

11.The expression format of Figure numbers should be unified to enhance the standardization of article format. For example, when referring to Figure 1 in the manuscript, it was expressed as "Fig 1", while when referring to other figures, some of them were expressed in different forms, such as "Figure 1" and "Fig. 1".

12.In the description of experimental results and discussion, the technical terms are not consistent enough for the expression of some similar concepts or indicators. For example, when describing nitrogen compounds, "ammonium", "ammonia nitrogen", "NH4+-N" and other expressions are used alternately, although the meaning is similar, but in the same article should try to uniform terminology.

13.Although repeated experiments on data accuracy and reliability were mentioned in the paper, the statistical analysis method used was not clearly explained in the comparison and analysis of multiple experimental results. For example, when comparing data such as differences in nitrogen removal rates and drug resistance among different strains, it was not mentioned what statistical tests (such as T-test, ANOVA, etc.) were used and what significance level was set.

6. PLOS authors have the option to publish the peer review history of their article (what does this mean? ). If published, this will include your full peer review and any attached files.

**Do you want your identity to be public for this peer review?** For information about this choice, including consent withdrawal, please see our Privacy Policy .

Reviewer #1: No

Reviewer #2: No

Reviewer #3: No

---

## [Author Response · Author response to Decision Letter 0]

16 Jan 2025

Upload the review as an attachment.

---

## [Decision Letter · Decision Letter 1]

31 Jan 2025

The nitrogen removal characterization and ecological risk assessment of Bacillus sp. isolated from mariculture systems in China with spatiotemporal difference

PONE-D-24-56326R1

Dear Dr. Yu,

We’re pleased to inform you that your manuscript has been judged scientifically suitable for publication and will be formally accepted for publication once it meets all outstanding technical requirements.

Kind regards,

Xingbin Sun

Academic Editor

PLOS ONE

Additional Editor Comments (optional):

Reviewers' comments:

Reviewer's Responses to Questions

**Comments to the Author**

1. If the authors have adequately addressed your comments raised in a previous round of review and you feel that this manuscript is now acceptable for publication, you may indicate that here to bypass the “Comments to the Author” section, enter your conflict of interest statement in the “Confidential to Editor” section, and submit your "Accept" recommendation.

Reviewer #1: All comments have been addressed

Reviewer #3: All comments have been addressed

2. Is the manuscript technically sound, and do the data support the conclusions?

Reviewer #1: Yes

Reviewer #3: Yes

3. Has the statistical analysis been performed appropriately and rigorously? 

Reviewer #1: Yes

Reviewer #3: Yes

4. Have the authors made all data underlying the findings in their manuscript fully available?

Reviewer #1: Yes

Reviewer #3: Yes

5. Is the manuscript presented in an intelligible fashion and written in standard English?

Reviewer #1: Yes

Reviewer #3: Yes

6. Review Comments to the Author

Reviewer #1: (No Response)

Reviewer #3: The author has made serious responses and modifications to my suggestions. Most of the modifications are targeted and the quality of the paper has been improved.

7. PLOS authors have the option to publish the peer review history of their article (what does this mean? ). If published, this will include your full peer review and any attached files.

**Do you want your identity to be public for this peer review?** For information about this choice, including consent withdrawal, please see our Privacy Policy .

Reviewer #1: No

Reviewer #3: **Yes: ** Jianyang Song

---

## [Editor Report · Acceptance letter]

PONE-D-24-56326R1

PLOS ONE

Dear Dr. Yu,

I'm pleased to inform you that your manuscript has been deemed suitable for publication in PLOS ONE. Congratulations! Your manuscript is now being handed over to our production team.

Kind regards,

on behalf of

Dr. Xingbin Sun

Academic Editor

PLOS ONE